# EmbedLLM: Learning Compact Representations of Large Language Models

**Richard Zhuang**[*] **Tianhao Wu**[*] **Zhaojin Wen** **Andrew Li**
**Jiantao Jiao** **Kannan Ramchandran**

University of California, Berkeley

## Abstract

With hundreds of thousands of language models available on Huggingface today, efficiently evaluating and utilizing these models across various downstream tasks has become increasingly critical. Many existing methods repeatedly learn task-specific representations of Large Language Models (LLMs), which leads to inefficiencies in both time and computational resources. To address this, we propose EmbedLLM, a framework designed to learn compact vector representations of LLMs that facilitate downstream applications involving many models, such as model routing. We introduce an encoder-decoder approach for learning such embeddings, along with a systematic framework to evaluate their effectiveness. Empirical results show that EmbedLLM outperforms prior methods in model routing both in accuracy and latency. Additionally, we demonstrate that our method can forecast a model's performance on multiple benchmarks, without incurring additional inference cost. Extensive probing experiments validate that the learned embeddings capture key model characteristics, *e.g.* whether the model is specialized for coding tasks, even without being explicitly trained on them. We open source our dataset, code and embedder to facilitate further research and application: `https://github.com/richardzhuang0412/EmbedLLM`.

## 1 Introduction

Recent breakthroughs in Large Language Models (LLMs) (Vaswani et al., 2023) have led to the creation of a vast array of models, each tailored for different use cases. These models, ranging from small, specialized models to large, general-purpose systems (Hao et al., 2022), differ significantly in their architecture, size, training data, and performance characteristics. For example, while some models excel as conversational agents, others may be more suitable for code generation or logical reasoning tasks. However, with this explosion of diverse LLMs comes a major challenge:

*How to efficiently manage, compare, and utilize the growing number of LLMs?*

Traditionally, benchmarking has served as the primary method for comparing LLMs, where each model is evaluated on a fixed set of test cases, and a score is generated to represent its performance. Meanwhile, model routing systems are developed to efficiently select models given queries of different types. An example workflow of these tasks can be seen in Figure 2 and Figure 3. While these approaches are often robust indicators of a model's strengths and weaknesses, the construction of their workflows induces repeatedly learning representations of various LLMs to suit each individual downstream tasks and is therefore time-consuming and compute-demanding.

In response to these challenges, we introduce **EmbedLLM**, a compute-friendly framework designed to learn compact vector representations of large language models that facilitates different tasks. EmbedLLM map models into a latent vector space that captures important model characteristics. More importantly, EmbedLLM produces a unified representation that can be simultaneously applied to various downstream tasks such as correctness forecasting (Section 5.1), model routing (Section 5.2), and benchmark accuracy evaluation (Section 5.3). The core idea is to enforce this representation

---

[*]Equal contribution.

learning through a reconstruction-based system that tries to predict the model's answer (correctness) from the learned embeddings, ensuring that each model's embedding retains the most salient features relevant to enhance performance across multiple scenarios.

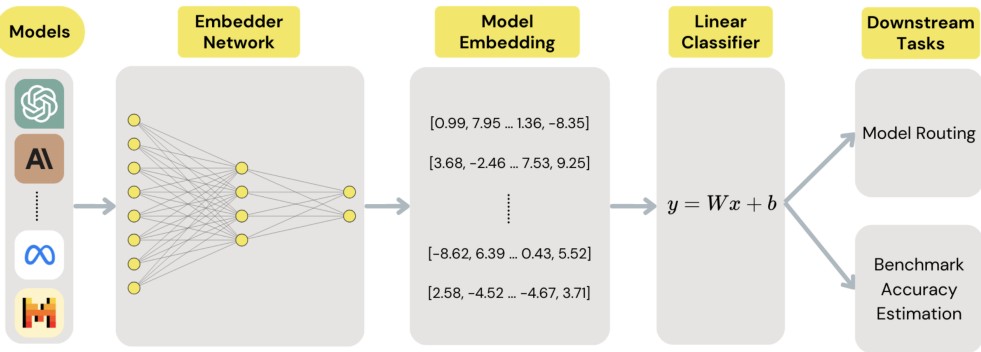

Figure 1: An illustration of the EmbedLLM Pipeline. An embedder network is pretrained from sample question-answer pairs from a pool of LLMs to map them into vector embeddings. Downstream applications like model routing are adapted by training an additional linear layer on top of these embeddings.

In summary, our contributions are as follows:

1. We propose a novel framework based on a reconstruction system for learning compact, low-dimensional representations of LLMs. To the best of our knowledge, this is the first work that explicitly introduces the concept of turning LLMs into embeddings.

2. We introduce an evaluation setup that employs the model embeddings to *simultaneously* predict model answer correctness on unseen questions, perform model routing, and predict benchmark accuracy with the addition of only a linear classifier. Our system demonstrates the potential for significantly reducing the need for task-specific re-training.

3. We perform probing experiments to validate that the learned embeddings capture meaningful information. We discover that models with similar characteristics remain close in the embedding space, and that the effect of incorporating each benchmark is reflected through the change in model embeddings.

EmbedLLM offers a scalable and unified approach to model evaluation and selection. By producing embeddings that encapsulate important features across tasks, our framework provides a versatile method to navigate the increasingly complex landscape of large language models.

## 2 RELATED WORK

**Representation Learning** There have been numerous attempts to learn representations of various types of information. For natural languages, Mikolov et al. (2013a) and Pennington et al. (2014) revolutionized the way models capture word semantics. In the field of computer vision, self-supervised techniques (Noroozi & Favaro, 2017) (Vondrick et al., 2018) are designed to learn low-dimensional representations that bolster downstream classification or segmentation performances. Inspired by these work and realizing an increasing demand of various LLMs being trained, we propose a creative framework to learn embeddings of LLMs.

**LLM Benchmarking** Benchmarking has been a standard way to compare LLMs, where a collection of questions/prompts is input to the LLMs and the quality of the corresponding responses is evaluated. However, with enormous inference cost incurred, current benchmarks typically aggregate model answers to a single accuracy metric, losing valuable insights from the diverse model responses; our work explores repurposing these inference results to gain a deeper understanding of model capabilities.

**LLM Routing** Our work focuses on predictive routing, which is a technique aimed at proposing the most suitable model given a task, without actually passing the query through each one of them. As

summarized in Hu et al. (2024), most routers adopt either a supervised technique (Ong et al., 2024; Shnitzer et al., 2023) or a reward-based method (Lu et al., 2023). Trained by seeing responses from different models to the same prompt, these systems are intrinsically building an understanding of key model characteristics. Our work establishes yet another new interesting and efficient research direction as we find model embeddings strengthen routing performances.

## 3 FORMULATION

### 3.1 PROBLEM SETUP

Let $\mathcal{M} = \{M_1, M_2 \cdots M_n\}$ be a set of different LLMs, $\mathcal{P}$ denote the set of all possible prompts, and $\mathcal{A}$ denote the corresponding set of possible answers. We can simply identify any LLM $M$ with an inference function mapping from a prompt space to an answer space $f_M : \mathcal{P} \to \mathcal{A}$, which outputs an answer $a \in \mathcal{A}$ given a prompt $p \in \mathcal{P}$.

Among downstream tasks, representations of different LLMs needed to be constructed in various ways. A naive example is benchmarking: Where the crucial part is to select a test prompt set $\mathbb{P}_{Bench} = \{p_1, p_2 \cdots p_m\}$ as well as an scoring function $g_{eval} : \mathcal{P} \times \mathcal{A} \to \mathbb{R}$, mapping model responses to a scalar score. Take MMLU as an example, each model is queried by a set of multiple-choice questions with four choices. The model's response is recorded by comparing the output probabilities of the answer choices "A", "B", "C", and "D". Then the responses are simply scored by matching with the ground truth answer key. Within this process, every model $M$'s behavior is essentially summarized by their output on a set of test questions$\{(p_1, a_{(M,1)}), (p_2, a_{(M,2)}) \cdots (p_m, a_{(M,m)})\}$.

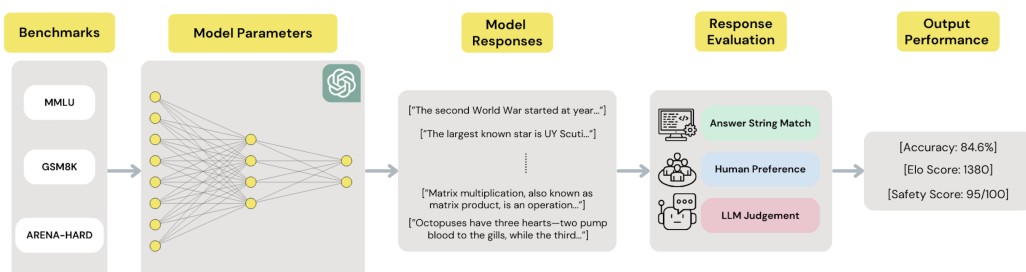

Figure 2: An illustration of the traditional workflow of LLM benchmarking.

Another example is model routing: given a pool of $n$ LLMs, a router function is defined in Ong et al. (2024) as a $n$-way classifier that assigns models to different queries to maximize response quality while minimize inference cost. Similarly, training such router also often involves in transforming each model into a relatively low-dimensional representation by utilizing sample question-answer pairs.

Identifying a common need for model characterization, we raise the question: what if we could accomplish the above tasks by **directly working with a unified, compact representation of LLMs**? In this work, we provide a framework to learn such a representation. We define an embedding function $\phi : \mathcal{M} \to \mathbb{R}^d$, parametrized by a neural network or otherwise, that maps each model $M$ to a compact vector representation in a latent embedding space. We aim to learn model embeddings that contain important features of LLMs that are useful to both quantifying differences between various models and aiding across downstream tasks.

### 3.2 EVALUATION METRICS

Evaluating the quality of model embeddings is crucial to ensure they effectively capture the underlying structure and semantics of the data which we care about for potential downstream tasks. The core idea is to use the embeddings to predict model behavior on unseen tasks by training an inference function $\psi : \mathbb{R}^d \times \mathcal{P} \to \mathbb{Q}$, that leverages a model embedding and new test prompts to predict a model's performances as quantified by a desired evaluation metric in the space $\mathbb{Q}$. For instance, if

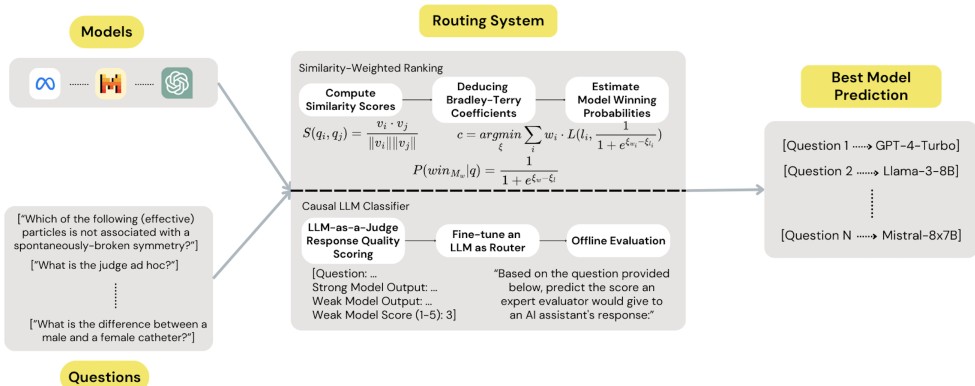

Figure 3: An illustration of the traditional workflow of model routing, using exemplar routing methodologies from Ong et al. (2024)

our task is to predict whether a model can correctly answer an unseen question, the inference function would be a network that takes in model embeddings and the unseen question (usually present in the form of its embedding as well) and output a binary label as the prediction of model correctness. Note that model responses can be evaluated through many means such as determining correctness (Hendrycks et al., 2021), judging by human (Chiang et al., 2024) or stronger LLMs (Zheng et al., 2023), or measuring any other task-specific metric. Hence, each task constitute to its own unit of measure and correspondingly determines an evaluation metric. In this work, we focus on evaluating on the following downstream tasks:

- **Correctness Forecasting:** We query the models on benchmarks equipped with ground truth answers, and produce binary correctness label for every model-question pair. For this task, a natural metric to use would be test prediction accuracy.

- **Model Routing:** With the help of a embedding vector of each model, we're able to develop a simple linear router that directly determine the model to be routed to using the probability of the answer correctness along with some learnable threshold. We measure the routing performance by reporting the average response quality[1], namely the proportion of times a router successfully route to an LLM that correctly answers the given query.

- **Benchmark Accuracy Prediction:** We treat the embeddings as primary features to train a linear model in predicting accuracies on unseen benchmarks (measured from 0% to 100% as a percentage). The metric used for this task would be the classical mean-squared-error (MSE) for linear regression.

In addition, we could directly compare between embeddings in their raw vector form following Mikolov et al. (2013b). For instance, models with similar traits should have embeddings that are closer in L2 distance.

## 4 EMBEDLLM

### 4.1 METHODOLOGY

In order to learn such embeddings, we draw inspirations from image reconstruction algorithms (He et al., 2022; Ronneberger et al., 2015): we want to learn a "reconstruction" system, where the choice of the reconstruction target is arbitrary and can be task-dependent. More concretely, let $\mathbb{Q}$ be a space of model performance metric, and $\mathcal{M}$ be a set of possible LLMs, $m$ be the number of models, $n$ be the number of questions/prompts. We want to learn a reconstruction network $\mathcal{R} : \mathbb{Q}^{m \times n} \to \mathbb{Q}^{m \times n}$

---

[1]Note that this metric can be insufficient under a regular routing setting (Ong et al., 2024) as it does not consider model cost, nevertheless it fits well to our goal of assessing whether our model embeddings effectively captures the strengths and weaknesses under the context of correctness forecasting.

that optimizes to reconstruct a matrix $X \in \mathbb{Q}^{m \times n}$, where $X_{ij} \in \mathbb{Q}$ denotes the model performance metric for model i on question j. The encoder-decoder architecture ensures the imposure of such constraint and enforce the model embeddings to efficiently capture the key characteristics of each model.

In this work, we decide to use the task of predicting model answer correctness as our auxiliary target, $i.e.$, $\mathbb{Q} = \{0, 1\}$. Notice that our training objective is only a decoy - any downstream task that requires understanding of model characteristics would qualify and our ultimate goal is to enforce the reconstruction network to learn a compact yet information-rich representation of the models in this process.

## 4.2 DATASET

We now describe the data collection process of correctness results of various LLMs' responses to questions from mainstream benchmarks.

We selected 112 open-sourced models[2] of various sizes, with both general purpose LLMs (Zhu et al., 2024) and specialized LLMs included to ensure comprehensive coverage. Then we aggregated responses of every model to 36,054 questions from the test sets of MMLU (Hendrycks et al., 2021), TruthfulQA (Lin et al., 2022) , SocialQA (Sap et al., 2019), PIQA(Bisk et al., 2019), MedMCQA(Pal et al., 2022), MathQA(Amini et al., 2019), LogiQA(Liu et al., 2020), GSM8K(Cobbe et al., 2021), GPQA(Rein et al., 2023), and ASDiv(Miao et al., 2020). The responses to these questions were acquired and evaluated through using the "lm-evaluation-harness" package (Gao et al., 2023) to give a binary correctness label for each model-question pair. We performed a random 80%-10%-10% train-validation-test split on the questions and used the sentence transformer "all-mpnet-base-v2" (Reimers & Gurevych, 2019) to convert the questions into an initial embedding state of dimension $dim_q = 768$. Consequently, our question embedding tensor $X$ has the shape (36054, 768) where the $i$-th row $X_i \in \mathbb{R}^{dim_q}$ is the embedding for model $i$ , and our label tensor is essentially a correctness matrix $Y$ with shape (112, 36054), where the $ij$-th entry $Y_{ij}$ represents the binary correctness of model $i$ answering question $j$.

## 4.3 ALGORITHM

We adopt an encoder-decoder architecture to learn model representations:

**Encoder:** The encoder consists of a model embedding network and a question embedding network. Let $dim_{embed}$ be the desired dimension of the model embedding. The model embedding network $\phi_m : \mathcal{M}^3 \to \mathbb{R}^{dim_{embed}}$ maps each model into a latent representation. Similarly, the question embedding network $\phi_q : \mathcal{P} \to \mathbb{R}^{dim_{embed}}$ maps each questions into a latent representation. In our setting, the question embedding network is a two-step transformation $\phi_q = g_{st} \circ h_{proj}$ where $g_{st} : \mathbb{P} \to \mathbb{R}^{dim_q}$ denotes the pre-processing (performed in Section 4.2) that turns each question from text to an initial embedding space by using sentence transformer, and $h_{proj} : \mathbb{R}^{dim_q} \to \mathbb{R}^{dim_{embed}}$ is a projection layer from this original embedding space to the same space as the model embeddings.

**Decoder:** The decoder is a binary classifier $\psi : \mathbb{R}^{dim_{embed}} \times \mathbb{R}^{dim_{embed}} \to \{0, 1\}$ that takes in both the encoded embeddings of the model and the question, and output whether the model answer the question correctly. For this work, our decoder is represented as $\psi(v_m, v_q') = \varphi(v_m \odot v_q')$ where $\varphi : \mathbb{R}^{dim_{embed}} \to \mathbb{R}^2$ is a linear classifier and $u \odot v$ represents the Hadamard (element-wise) product between two vectors. For each model-question pair, this decoder network outputs two logits $p_{(m,q)_0}$ and $p_{(m,q)_1}$, and the "correctness score" $s_{m,q} = \sigma(p_{(m,q)_1} - p_{(m,q)_0})$ represent the predicted probability of the model $m$ correctly answering question $q$, where $\sigma(x)$ is the sigmoid function.

Suppose $y$ is the correctness label of model $m$ answering question $q$, we calculate the following BCE loss function during training,

$$L(m, q, y) = - (y \cdot \log(s_{m,q}) + (1 - y) \cdot \log(1 - s_{m,q})) \tag{1}$$

In essence, this algorithm analogizes to a **matrix factorization** algorithm, where we learn a $n \times m$ model embedding matrix and a $m \times p$ question embedding matrix such that their product recon-

---

[2]See appendix for full list.

[3]In the actual implementation, we give each model an index as an identification in $\mathcal{M}$.

structs the original $n \times p$ correctness matrix. In the following sections, we refer to this algorithm as EmbedLLM.

## 5 EXPERIMENT RESULTS

As described in Section 3.2, we conducted experiment in correctness forecasting, model routing, and benchmark accuracy prediction to evaluate the quality of the learned embeddings.

### 5.1 CORRECTNESS FORECASTING

For correction prediction, we compare the effect of EmbedLLM to a K-Nearest-Neighbor classifier (Fix, 1985). In the context of our formulation, although without an explicit embeddings for models, a KNN-classifier can be seen as using the integration of all question-correctness tuples from a model as its "embedding", and making inference from this aggregation. Specifically, given a question and a model, the classifier outputs the majority vote of whether the model correctly predicts the nearest neighbor questions. For brevity, we refer to this approach as KNN in the subsequent text. As mentioned in Section 3.2, we use correctness forecasting accuracy on the test set as the evaluation metric.

We evaluate the performance of KNN and EmbedLLM across various sizes of training set. We produce the smaller training sets from randomly subsetting from the full training set. For each training set, we conduct hyperparameter tuning (number of neighbors for KNN, model embedding dimension for EmbedLLM) on a fixed validation set and evaluate prediction accuracy using a fixed test set. The result in Table 1 indicates a better scalability of our method.

| Algorithm | Dataset Size | | | | | | |
|---|---|---|---|---|---|---|---|
| | 1K | 5K | 10K | 15K | 20K | 25K | Full (29K) |
| KNN | 0.6372 | 0.7078 | 0.7107 | 0.7128 | 0.7143 | 0.7146 | 0.7152 |
| EmbedLLM | 0.6443 | 0.7331 | 0.7362 | 0.7378 | 0.7390 | 0.7394 | **0.7409** |

Table 1: Performance of predicting model correctness on a fixed unseen test set. The columns indicate the number of questions in the training set. EmbedLLM constantly outperforms KNN on training sets of all scale. We see a steady increase in the performance as the dataset size grows, indicating further scalability of our method.

### 5.2 MODEL ROUTING

Using the same correctness data, we can evaluate the quality of using EmbedLLM as router. For this task, we evaluate the router's accuracy by measuring the proportion of times it successfully route to a model that could correctly answer the given query. For a test question $q_k$, we pass it through the router network along with all $n$ possible model embeddings, producing $n$ correctness score $s_{M_1,q_k}, s_{M_2,q_k}, \cdots, s_{M_n,q_k}$, and check if the model with the highest correctness score correctly answers the question. Aggregating through all questions, we report the router accuracy as $acc_{router} = \frac{1}{m} \sum_{k=1}^{m} \mathbb{1}\{X_{i^*,k} = 1\}$ where $i^* = \arg\max\{s_{M_i,q_k} | i = 1 \cdots n\}$ is the routed model and $X$ is the correctness matrix described in Section 4.2.

We compare the performance of the EmbedLLM router with two baselines. The first one is the single-best model router which always selects the model with the highest accuracy and thus gives a constant accuracy. The second one is a random router that select each model the same number of times as the EmbedLLM router, but instead randomly assign models to questions. For this router, we can calculate its expected accuracy given the proportions of times each model is selected by the EmbedLLM router. For instance, if our router selects $M_1$ 70% of the time, $M_2$ 20% of the time, and $M_3$ 10% of the time, the expected accuracy of the random router will be calculated as a weighted accuracy $acc_{weighted} = 0.7 * acc_{M_1} + 0.2 * acc_{M_2} + 0.1 * acc_{M_3}$. Note that this weighted accuracy will always be smaller than the single-best model accuracy - we propose this metric as an evaluation of how well our router direct models to the questions they are good at given a fixed "budget" of model calls. We report both the overall accuracy across the whole test set and accuracy per source benchmark.

As seen in Figure 4, EmbedLLM router performs better than both the single-best router and the random router overall. As the best performing model is re-determined on every benchmark, the single-best model router performs better for each benchmark than in overall case. Here, EmbedLLM router achieves near single-best model router accuracy while still managing to utilize the respective strengths of different models which is shown by the significant difference between accuracies of EmbedLLM router and weighted router.

Another advantage of EmbedLLM router we find is its high routing speed. On one NVIDIA A100 80GB GPU, it takes in average 3.80 seconds for EmbedLLM router to route 3,000 questions on 50 repeated trials which is basically free compare to the downstream model inference time. Specifically, compared to the causal LLM router in Ong et al. (2024) that processes less than 50 requests per second, our router delivers more than 750 model selections, which is **15x faster** while selecting from a much larger model pool (112 models against 2 models for causal LLM router). In addition, it only takes less than 1GB of peak GPU memory when training a EmbedLLM router using our dataset, which is **60x cheaper** than fine-tuning Llama-3-8B as a router in terms of memory usage. This illustrates the superiority of EmbedLLM router in both performance, latency, and training cost.

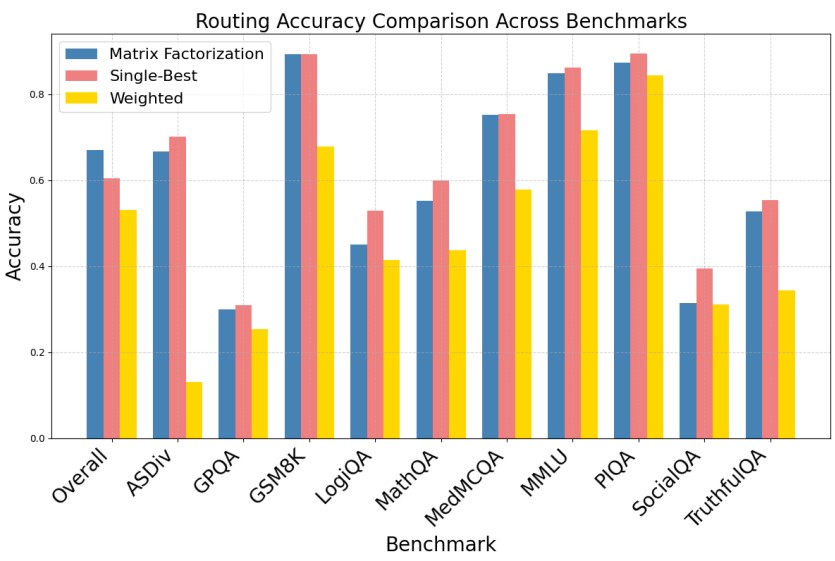

Figure 4: Performance accuracy of EmbedLLM router compared to baselines. EmbedLLM router performs better across the whole test set and achieves accuracies close to the single-best model on every benchmark.

## 5.3 BENCHMARK ACCURACY PREDICTION

To predict model's average accuracy on a benchmark $B$, we trained EmbedLLM using leave-one-out correctness data, which includes correctness results of all models on all questions except the ones in $B$. Then we take the model embeddings directly as features to train a linear regression of the form:

$$a\mathbf{E} = \mathbf{y}$$

where $E$ is a model embedding matrix with the $i$-th row representing the model embedding for the $i$-th model, and the $j$-th entry in the vector $\mathbf{y}$ corresponds to the $j$-th model's average correctness accuracy on the test benchmark, which is a number from 0 to 1.

For each test benchmark, we conducted 100 random train-test splits on the 112 models contained in our dataset, trained a linear regression on the training set, and evaluated the correlation between model embeddings and test benchmark performances on the test set through applying Kendall's Tau test[4]. From Section 5.3, statistical significance is found in 7 out of the 10 benchmarks, indicating

---

[4]The Kendall's Tau test is a measure of correspondence between two rankings. We use this test to see if model ability can be correctly ordered simply using a linear system with the embeddings as the only feature.

that model embedding contains information to distinguish between model performances on most benchmarks.

| Benchmark | Significance |
|-----------|-------------:|
| MathQA | 100 |
| LogiQA | 100 |
| MedMCQA | 100 |
| PIQA | 98 |
| TruthfulQA | 96 |
| MMLU | 94 |
| GSM8K | 93 |
| GPQA | 10 |
| ASDiv | 6 |
| SocialQA | 3 |

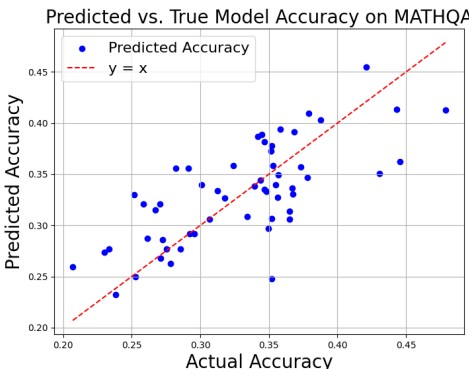

Figure 5: **Left:** Sorted Kendall's Tau test result of accuracy prediction on the benchmarks. The "Significance" column represents the number of times with significant correlation detected (at a 5% significance level) out of 100 random model splits. **Right:** An example comparing actual model accuracies on MathQA against model accuracies on MathQA predicted from the embeddings trained without MathQA data.

Notice that this prediction systems works even when we leave out large benchmarks like MMLU, as our method predicts MMLU accuracy to a statistically significant extent. As number of models, model sizes and number of benchmarks are still rising rapidly, enabling benchmark accuracy prediction through model embeddings is vital to save both time and compute from repeatedly inferencing models on every new benchmark with potentially overlapping questions from previous ones.

## 6   WHAT INFORMATION IS IN THE MODEL EMBEDDINGS

In this section we describe the probing experiments designed to understand what information is captured in the embedding.

### 6.1   SANITY CHECK USING SIMILARITY

We expect the model embeddings to satisfy some basic properties: If two models $M, M'$ generate the same answers for every prompt, then their embeddings are the same. Similarly, models with similar characteristics, trained using similar data, or adopted similar training pipelines should have similar embeddings, and vice versa. For instance, the model embedding of DeepSeekMath-7B (Shao et al., 2024) should be more similar to the embedding of other math models like MetaMath-Llemma-7B (Yu et al., 2023) than to the embedding of Medicine-LLM-13B (Cheng et al., 2024) which is adapted for biomedical applications. This property is easily fulfilled by EmbedLLM as any two identical/similar models of such would produce identical/similar correctness result against most questions.

As a further sanity check, we assign binary labels to the 112 models we have evaluated according to the following 6 keywords: [7B, 13B, 70B, Coding, Bio/Med, Physics], forming 6 characteristic communities. For each community, we compare between the average intra-community and inter-community L2 distance of the embeddings. As shown in Figure 6, for all above 6 communities, the averaged intra-community L2 distances are smaller than the inter-community ones. This provides a preliminary guarantee that our embeddings are "meaningful" with respect to distance metrics.

### 6.2   EMBEDDINGS CAPTURE INTRINSIC CHARACTERISTICS OF BENCHMARKS

Next, as indicated from Section 5.3, as a set of model embeddings is produced from a fixed training set, the embeddings seem to capture information of some benchmarks better and overlook information in some benchmarks. Hence, we design a set of ablation experiments to further understand the contribution of each benchmarks in the training data. Specifically, extend-

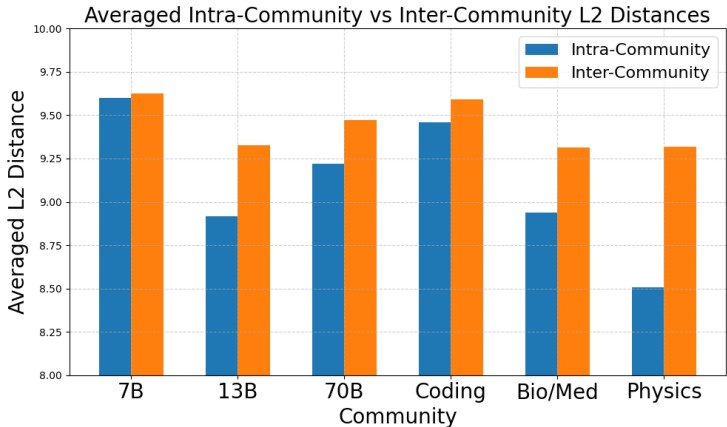

Figure 6: The averaged intra-community L2 distance of the model embeddings is closer for all 6 communities selected, suggesting that basic model traits are captured in the latent embedding space.

ing the experiment setup from Section 5.3,we have a question embedding tensor $X$ of shape (num_questions, embedding_dimension) and a label tensor $Y$ where $Y_{ij}$ is the binary label of whether model $i$ correctly answers question $j$, with questions from a set of benchmarks $S = \{B_1, B_2 \cdots B_n\}$, to measure the effect of incorporating/removing an "contributor" benchmark $B_i$ on predicting correctness of a "testee" benchmark $B_j$, we:

1. Construct two sets of benchmarks $S_{added} = S \setminus B_j$ and $S_{removed} = S \setminus (B_i \cup B_j)$ and produce two new sets of question embedding and label tensor, $X_{added}$, $X_{removed}$, $Y_{added}$, and $Y_{removed}$, so that only questions containing in $S_{added}$ and $S_{removed}$ are kept respectively.

2. Train a EmbedLLM embedder separately on $(X_{added}, Y_{added})$ and $(X_{removed}, Y_{removed})$ to get two sets of model embeddings $E_{added}$ and $E_{removed}$, and respectively perform zero-shot benchmark prediction on $B_j$ with 100 random splits of models as in Section 5.3. Aggregate the total test mean squared error (MSE) $e_{removed}$ and $e_{added}$.

3. Take the difference between the two error to compute a contribution score $C_{ij} = e_{removed} - e_{added}$ which quantifies the improvement on predicting model accuracy on $B_j$ when $B_i$ is added in training.

Essentially, we hypothesize that the addition/removal of every training benchmark would be reflected through the change in model embeddings, which then induces a performance difference in benchmark accuracy prediction. With this setup, we produce a $n \times n$ ($n$ is the total number of benchmarks) contribution matrix $C$ where the where the $ij$-th entry can be extracted exactly as $C_{ij}$ from the steps above[5]. To aggregate the overall effect of one specific benchmark $B_i$ as the contributor benchmark, we compute a row sum of the contribution matrix $\sum_j C_{ij}$. This can be interpreted as the total improvements of adding benchmark $B_i$ on predicting model correctness of the rest of the benchmarks. Correspondingly, the column sum $\sum_i C_{ij}$ represents the total amount of improvement of predicting correctness of benchmark $B_j$ when each of the rest of the benchmarks is added.

Some noticeable phenomenon emerged from this set of experiments[6]:

1. We find that incorporating MMLU into training the embeddings significantly help predicting accuracies on other benchmarks. This result matches the comprehensive nature of MMLU as it contains questions covering various topics.

---

[5]The diagonal entries of this matrix are set default to 0 as it is meaningless to train and test on the same benchmark.

[6]We omit results of SocialQA in probing experiments as most of our models perform similarly poorly, resulting in all model accuracies crowding in the low accuracy region and becoming indistinguishable.

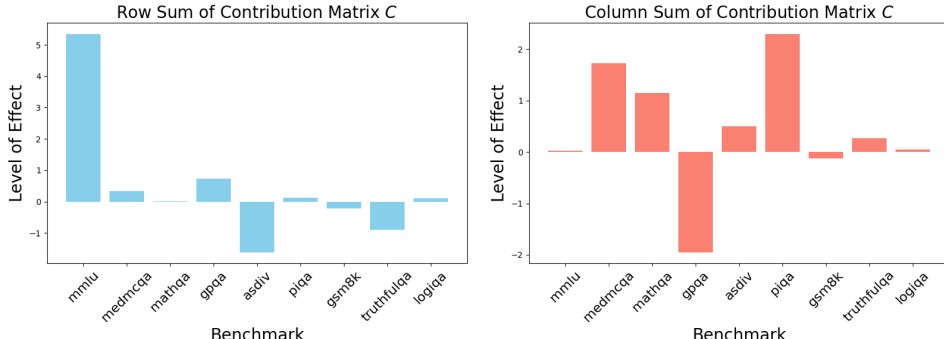

Figure 7: **Left:** Effect of removing benchmarks on testing all other benchmarks. Higher value suggests that the addition of that benchmark into the training of model embedding enhances predicting model correctness on the rest of the benchmarks. **Right:** Effect of removing each of the rest of the all other benchmarks on the benchmark being tested. Higher value suggests that predicting models' correctness from the tested benchmark becomes easier when other benchmarks are added into training.

2. We find that incorporating other benchmarks into training set would harm the embedding's predictive power on GPQA. This suggests that additional information that the embeddings capture from incorporating new benchmarks into training set is unrelated or negatively related to model performance on GPQA. In fact, GPQA is an extremely difficult benchmark for current LLMs, so this finding aligns with our expectation as model's ability in answering simpler questions clearly do not transfer to answering harder ones.

3. Additionally, we identify subsets of benchmarks that mutually improve each other's accuracies when incorporated into training. For instance, there is a total MSE improvement of 0.271 when GSM8k is incorporated to predicting MathQA, 0.190 when MathQA is incorporated to predict ASDiv, and 0.103 when ASDiv is incorporated to predict MathQA. As all three benchmarks are math-related, we can deduce that the level of math knowledge of our selected models are indeed learned and reflected in our model embeddings.

## 7 LIMITATION

Despite the promising results, our work has several limitations. First, our dataset, though effective in demonstrating the potential of our embeddings with a limited number of samples, is relatively small. With data from only 112 models, the embeddings we extract are moderately sparse which limits deeper exploration of relationships between them. Second, EmbedLLM is a static system. Despite the low cost[7], introducing new models still requires retraining. Lastly, our study is restricted to correctness-based datasets, leaving other potentially valuable data types, such as text embeddings of model outputs, unexplored. To address these limitations, we have open-sourced our datasets and codebase for further research and experimentation.

## 8 CONCLUSION

We showcase the possibility of learning an unified, compact representation of LLMs. Through extensive empirical evaluation, our method displays solid performance on correctness forecasting, model routing, and benchmark accuracy prediction, while significantly reducing the need of retraining and avoiding repetitive evaluations. Furthermore, we conduct various probing experiment to understand the information contained in the model embeddings. The results show that our embeddings capture not only key characteristics of the models, but also properties of the data used to train the embedder.

---

[7]Training EmbedLLM on a correctness matrix of around 20,000 questions on 112 models for 50 epochs with batch size or 2,048 costs 107.71 TFlops, approximately equivalent to the querying a 7B model for 60 times using an input of length 128.

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

ACKNOWLEDGEMENTS

This work was partially supported by NSF Grants IIS-1901252 and CCF-2211209. We sincerely thank the funding agencies for their support. Additionally, we appreciate the valuable feedback from the reviewers that helps improve this work.

## A    APPENDIX

### A.1    MODEL LIST

Here is an exhaustive list of models that we extract our dataset from:

| | |
|---|---|
| meta-llama/LlamaGuard-7b | meta-llama/Llama-2-13b-chat-hf |
| 01-ai/Yi-34B-Chat | meta-llama/Llama-2-70b-chat-hf |
| WizardLM/WizardLM-70B-V1.0 | allenai/tulu-2-dpo-70b |
| lmsys/vicuna-13b-v1.5 | lmsys/vicuna-33b-v1.3 |
| Qwen/Qwen-14B-Chat | upstage/SOLAR-10.7B-Instruct-v1.0 |
| openchat/openchat-3.5-0106 | openchat/openchat-3.5 |
| berkeley-nest/Starling-LM-7B-alpha | HuggingFaceH4/zephyr-7b-beta |
| TheBloke/tulu-30B-fp16 | mistralai/Mistral-7B-Instruct-v0.1 |
| tiiuae/falcon-40b-instruct | lmsys/vicuna-13b-v1.5-16k |
| codellama/CodeLlama-34b-Instruct-hf | TheBloke/WizardLM-13B-V1.2-GGUF |
| lmsys/vicuna-7b-v1.5 | NousResearch/Nous-Hermes-13b |
| project-baize/baize-v2-13b | lmsys/vicuna-7b-v1.5-16k |
| mosaicml/mpt-30b-instruct | meta-llama/Llama-2-7b-chat-hf |
| TheBloke/koala-13B-HF | nomic-ai/gpt4all-13b-snoozy |
| h2oai/h2ogpt-gm-oasst1-en-2048-open-llama-13b | mosaicml/mpt-7b-chat |
| databricks/dolly-v2-12b | stabilityai/stablelm-tuned-alpha-7b |
| OpenAssistant/oasst-sft-4-pythia-12b-epoch-3.5 | deepseek-ai/deepseek-llm-67b-chat |
| NousResearch/Nous-Hermes-2-Yi-34B | CausalLM/34b-beta |
| SUSTech/SUS-Chat-34B | SUSTech/SUS-Chat-72B |
| Qwen/Qwen-72B | Intel/neural-chat-7b-v3-3 |
| ibivibiv/alpaca-dragon-72b-v1 | JaeyeonKang/CCK-Asura-v1 |
| ConvexAI/Luminex-34B-v0.2 | ConvexAI/Luminex-34B-v0.1 |
| CorticalStack/pastiche-crown-clown-7b-dare-dpo | eren23/ogno-monarch-jaskier-merge-7b-OH-PREF-DPO |
| bardsai/jaskier-7b-dpo-v5.6 | FelixChao/Scorpio-7B |
| dfurman/HermesBagel-34B-v0.1 | kevin009/llamaRAGdrama |
| sail/Sailor-7B | AiMavenAi/Prometheus-1.3 |
| Q-bert/Optimus-7B | cognitivecomputations/yayi2-30b-llama |
| zhengr/MixTAO-7Bx2-MoE-v8.1 | fblgit/UNA-SimpleSmaug-34b-v1beta |
| mistralai/Mixtral-8x7B-Instruct-v0.1 | microsoft/Orca-2-13b |
| EleutherAI/pythia-12b | cloudyu/Mixtral-11Bx2-MoE-19B |
| rishiraj/CatPPT-base | Deci/DeciLM-7B |
| microsoft/phi-2 | scb10x/typhoon-7b |
| 01-ai/Yi-6B-200K | 01-ai/Yi-6B |
| TigerResearch/tigerbot-13b-base | augmxnt/shisa-base-7b-v1 |
| microsoft/phi-1.5 | golaxy/gowizardlm |
| bigscience/bloom-7b1 | mlabonne/AlphaMonarch-7B |
| CultriX/NeuralTrix-bf16 | shadowml/MBeagleX-7B |
| yam-peleg/Experiment26-7B | deepseek-ai/deepseek-math-7b-instruct |
| meta-math/MetaMath-Mistral-7B | kyujinpy/Sakura-SOLRCA-Math-Instruct-DPO-v1 |
| FelixChao/llama2-13b-math1.2 | Plaban81/Moe-4x7b-math-reason-code |
| MaziyarPanahi/WizardLM-Math-70B-v0.1 | abhishek/zephyr-beta-math |
| meta-math/MetaMath-Llemma-7B | EleutherAI/llemma-34b |
| EleutherAI/llemma-7b | FelixChao/vicuna-7B-physics |
| Harshvir/Llama-2-7B-physics | FelixChao/vicuna-7B-chemical |
| BioMistral/BioMistral-7B | BioMistral/BioMistral-7B-DARE |
| PharMolix/BioMedGPT-LM-7B | Biomimicry-AI/ANIMA-Nectar-v2 |
| codellama/CodeLlama-7b-hf | codellama/CodeLlama-13b-Instruct-hf |
| deepseek-ai/deepseek-coder-1.3b-base | deepseek-ai/deepseek-coder-6.7b-instruct |
| OpenBuddy/openbuddy-codellama2-34b-v11.1-bf16 | TheBloke/CodeLlama-70B-Instruct-AWQ |
| AdaptLLM/medicine-chat | AdaptLLM/medicine-LLM |
| AdaptLLM/medicine-LLM-13B | Writer/palmyra-med-20b |
| SciPhi/SciPhi-Self-RAG-Mistral-7B-32k | Neko-Institute-of-Science/metharme-7b |
| Neko-Institute-of-Science/pygmalion-7b | SciPhi/SciPhi-Mistral-7B-32k |
| shleeeee/mistral-ko-tech-science-v1 | codefuse-ai/CodeFuse-DeepSeek-33B |
| WizardLM/WizardCoder-Python-34B-V1.0 | bigcode/octocoder |
| meta-llama/Meta-Llama-3-8B | meta-llama/Meta-Llama-3-8B-Instruct |
| meta-llama/Meta-Llama-3-70B | meta-llama/Meta-Llama-3-70B-Instruct |
| meta-llama/Meta-Llama-Guard-2-8B | Qwen/Qwen1.5-32B-Chat |
| Qwen/Qwen1.5-4B-Chat | Qwen/Qwen1.5-0.5B-Chat |
| Qwen/Qwen1.5-7B-Chat | Nexusflow/Starling-LM-7B-beta |
| google/gemma-7b-it | google/gemma-2b-it |

Table 2: The comprehensive list of the 112 models used to create the correctness dataset. This dataset is created on May 2025 so models released after that time are not available.

