# OpenReview forum: "EmbedLLM: Learning Compact Representations of Large Language Models"
_ICLR.cc/2025/Conference — ICLR 2025 Spotlight_

### Official Review · Reviewer_HPLR · 2024-11-02

**Soundness:** 3
**Presentation:** 3
**Contribution:** 3
**Rating:** 8
**Confidence:** 4

**Summary:**

The goal of this paper is to represent LLMs using an embedding that can predict performance on new inputs and benchmarks. Such an embedding can be learned from matrix factorization methods applied to a matrix that contain the behavior of the LLM on several data points as well as behavior of other LLMs on these datapoints, i.e., a matrix with rows as LLMs and columns as data points, and each cell indicate an LLM's performance on a specific data point. The factorization aims to reconstruct this matrix by learning an embedding for each LLM and an embedding for each data point (built from its static embedding which is further projected into the same embedding space of LLM).

The resulting embeddings are evaluated to predict performance of new data points and benchmarks and to improve model routing.

**Strengths:**

1. A simple and intuitive method.
2. Once the embedding of an LLM is built, the performance of the LLM can be accessed without access to the model.

**Weaknesses:**

1. The paper can be motivated better and clarify why the current formulation is intuitive, i.e., a model's behavior on new data points can be predicted based on its behavior on already existing data points.
2. Details are hard to follow or underspecified, e.g., kNN classifier, random routing.
3. There are no references to highly similar work that predicts performance on a new task based on the performance of existing tasks. For example, Xia et al. Predicting Performance for Natural Language Processing Tasks, 2020
4. Calling the method as encoder-decoder in Sec 4.3 is confusing. The decoder is nothing but a classifier, and calling it straightforwardly that will make it easier to follow instead of calling it a decoder.
5. There is no analysis/visualations of the embeddings. Current histogram plot in Figure 6 is not that informative. Could you provide TSNE plots of the resulting embeddings.

**Questions:**

1. Why does the method assume that answer have to be a binary label? Is this a limitation of this method?
2. The explanation of kNN-classifer baseline is unclear. How is final label chosen? -- based on the majority voting of k-nearest neighbours?
3. Lines 264--267 are unclear. What is mxp and nxp represent?
4. Some qualitative discussion with examples on what happens when the resulting data points are widely different from already seen data points will be useful.
5. Does the prompt embedding have an impact on the results, i.e., using better recent LLM-based embeddings.
6. I would also like to see the TSNE plots of the embeddings.

Minor:
1. Vaswani et al. should be 2017. The paper opens up with a wrong citation. This paper is also wrongly attributed to LLMs. You could be more careful with your citations.
2. The method should perhaps be named EmbedLLM rather than Matrix Factorization.

I am willing to increase the score based on better presentation and qualitative analysis. The paper is quite weak in this aspect.

====

I increased my score after author rebuttal. I am worried about the quality of these embeddings given that no patterns are observed in TSNE plots, but a score of 7 is appropriate for other clarifications. Since there is no 7, I had to chose 8.

---

> ### Author Response · Authors · 2024-11-21
> **Rebuttal Response (1/2)**
>
> We thank the reviewer for their comments and for taking the time to review our paper. We also appreciate their comments about finding our work intuitive.
>
> We respond to some of the reviewer’s comments below
>
> Q1: “Why does the method assume that answer have to be a binary label? Is this a limitation of this method?”
>
> **Our method does not need the label to be binary only**. Our emphasis is to convey that we can employ such encoder-decoder architecture to learn compact representations of various LLMs that facilitate downstream tasks, given some data of model behaviors. We choose a dataset of binary correctness labels because it is the most accessible feature that quantifies model behavior. Nonetheless, **our framework is certainly generalizable to learning from other formats of data**, for instance we could learn to reconstruct exact model responses to questions (of course in this setting the encoder and decoder needs to be more carefully designed and the scale of the training is larger).
>
> Q2: “The explanation of kNN-classifer baseline is unclear. How is final label chosen? -- based on the majority voting of k-nearest neighbours?”
>
> Yes, the KNN classifier aggregates the label by getting the majority vote of the K nearest neighbors where we performed hyperparameter tuning on the optimal number of neighbors.
>
> Q3: “What is mxp and nxp represent?”
>
> The correctness matrix has shape (n,p) where n is the number of models and p is the number of prompts(questions). As a result, m is the dimension of our model embeddings which is itself a hyperparameter and thus can be flexible. We thank the reviewer for pointing this out and we will update our writing with an additional explanation on what these symbols mean.

---

> ### Author Response · Authors · 2024-11-21
> **Rebuttal Response (2/2)**
>
> Q4: “Some qualitative discussion with examples on what happens when the resulting data points are widely different from already seen data points will be useful.”
>
> Some of our probing experiments serve for that purpose. For instance, in figure 5, a linear regressor put on top of our embedding, which is trained from all 10 benchmarks except SocialQA, does not exhibit statistical significance in predicting SocialQA performance.
>
> We would also like to clarify that our current embedder certainly does not capture ALL characteristics of the models. Intuitively, if the model behavior data has zero information about how models perform in complex reasoning tasks, then we argue that no method could effectively predict model behaviors on them. However, we would like to emphasize that the key contribution of our work is that **our framework could effectively capture and compress whatever information that is available in our dataset** in a compact embedding form, and our experiment in correctness forecasting as well as model routing suggests that the **model embeddings are powerful summarizers** of model strengths and weaknesses within the range of model behaviors that is tested in our selected benchmarks.
>
> Q5: “Does the prompt embedding have an impact on the results, i.e., using better recent LLM-based embeddings.”
>
> Specifically on our dataset, the choice of the embedder that produces the prompt embedding **does not affect the result to a large extent**. We provide some experimental detail below:
>
> For the experiments in the paper, we chose “sentence-transformers/all-mpnet-base-v2” as our embedder and on the full training set training from embeddings from this embedder gives 74% test accuracy. We further explored 4 other embedders from the MTEB leaderboard on HuggingFace: “jxm/cde-small-v1”, “nomic-ai/nomic-embed-text-v1-ablated”, “sentence-transformers/msmarco-bert-co-condensor”, and “sentence-transformers/LaBSE”. Using the same set of hyperparameters, we found that all of the training using these embedders gives a test accuracy ranging from 72% to 74%.
>
> Intuitively, since our questions are more like knowledge tests, we expect the embedder to learn the differences between question embeddings in the form like “which field is the question belonging to”. Nevertheless, we agree that a better embedder could result in better performance when the model behavior data become more intricate or when the label of the data becomes related to the actual model response.
>
> Q6: “I would also like to see the TSNE plots of the embeddings.”
>
> As mentioned in our limitation section, because of the limited number of model (112), the embeddings are often quite sparse and therefore dimension reduction techniques did not apply very well on the embeddings. However, we would like to reassure the reviewer that we conducted extensive probing experiments (Section 6) to ensure that the embeddings capture meaningful information. Specifically, the performance of our linear prober placed on top of the embeddings are **highly affected by changes in the training set and their changes align with the qualitative natures of the benchmark we selected**. We hope that these probing experiments could serve as an alternative to tSNE plots.
>
> On the Minor issues: We appreciate the reviewer for pointing out these details and we will certainly refine our writing in an updated version.
>
> Finally, we would like to thank the reviewer once more for their comments which we find to be really constructive. If the reviewer finds our responses above to answer their questions, we gently request the reviewer to take them into account before finalizing their reviews and revise their scores as they find appropriate.

---

> ### Comment · Reviewer_HPLR · 2024-11-26
>
> Thank you for your response! I still feel some important questions are unanswered.
>
> > Q6: I would also like to see the TSNE plots of the embeddings.
>
> I am puzzled by the response. Why is the resulting model embedding sparse? In Q3, you indicate "m" is quite flexible, and if so, is it not possible to build a dense embedding. Can you point me to the discussion in the paper where you describe what values of 'm' are used. But TSNE plots are based on distances, so I don't see sparsity as a problem, or is it? As I pointed in weakness 5, Figure 6 isn't that useful to give a wholistic view of model similarities.
>
> > Q5: “Does the prompt embedding have an impact on the results, i.e., using better recent LLM-based embeddings.”
>
> Here I meant LLM-based embeddings not MLM based embeddings. For example, LLM based embeddings like LLM2Vec or GritLM style. In your explanation, you imply that question category (domain) is more important than the difficulty of the question. That is intriguing because I thought the main usefulness of the work is at estimating whether a model can solve a given question but not how good it is in a domain? Isn't it?
>
> Could you also comment on weakness 3 and 4?

---

> ### Author Response · Authors · 2024-12-03
>
> Thanks for the further discussion! Here are responses to some of your questions:
>
> 1. About TSNE plots, upon further review, we agree that sparsity may not inherently preclude TSNE from being used effectively. In our case, TSNE plots of our embeddings were not sparse; however, the resulting plots do not reveal clear trends or clusters. The points appear more scattered. We acknowledge this as a current limitation. To improve this aspect, we plan to explore approaches such as training with a subset of more specialized models (e.g., only include models specialized in math and in medicine) to potentially create more distinguishable clusters. Additionally, adding more models to the dataset could address sparsity concerns and allow for clearer separations. We appreciate this suggestion and will address it in the final version of the paper to enhance the interpretability of our embeddings.
>
> 2. About LLM-based embedding, we appreciate this clarification and we will conduct more experiments on the effect of the embedder. We believe that more informative embedding could definitely enhance performances on downstream tasks and we will consider this enhancement for future work.
>
> About question category vs. difficulty, we believe that the behavior of our routing mechanism suggests that the embeddings are capturing characteristics aligned with question domains rather than question difficulty. Specifically, our MF router **utilizes a diverse set of models for different questions**, which indicates that the embeddings capture nuances about which models are better suited for specific domains. This behavior aligns with our classifier design—a linear classifier applied to the Hadamard product of the question and model embeddings—which encourages an alignment between model and question domains. If the embeddings were purely reflective of difficulty, we would **expect the router to favor a smaller subset of strong models across all questions**, which is not observed in our experiments. We will conduct more controlled experiments to validate this claim.
>
> 3. For weakness 3, we will incorporate references to relevant prior work, including Xia et al. (2020), in the final version of the paper. One thing worth mentioning is that while related works have focused on predicting performance on new tasks based on existing tasks, our primary focus is on compactly vectorizing model characteristics. Performance prediction serves as an auxiliary task within our broader framework, which is designed to adapt to any measurable model behavior data.
>
> 4. About weakness 4, we thank the reviewers for pointing out this possible confusion. In the specific setting of our binary correctness dataset, the decoder is indeed a binary classifier, and the reason why we call it a decoder is because it is used to reconstruct every entry of our correctness matrix. We will place more emphasis in explaining this in the updated writing to make it less ambiguous. Nonetheless, we re-emphasize that our framework is generalizable to learning from other formats of data, for instance we could learn to reconstruct exact model responses to questions and in that setting the decoder would be reconstructing text as the reviewer suggested. In this work, the main focus is to convey that we can employ such encoder-decoder architecture to learn compact representations of various LLMs that facilitate downstream tasks, given some data of model behaviors.
>
> Overall, we thank the reviewer again for their constructive and detailed comments. These insights have been invaluable in identifying areas for improvement, and we are committed to addressing them comprehensively in the final version.

---

### Official Review · Reviewer_Gpg4 · 2024-11-03

**Soundness:** 3
**Presentation:** 2
**Contribution:** 4
**Rating:** 8
**Confidence:** 4

**Summary:**

The paper presents EmbedLLM, a framework for creating compact vector embeddings of large language models (LLMs) to improve efficiency in tasks like model routing and benchmark performance prediction. EmbedLLM uses an encoder-decoder architecture to map LLMs into a unified embedding space that captures important model characteristics. This representation allows accurate model selection and performance forecasting across multiple tasks without repetitive re-evaluation, reducing both time and computational costs. Experiments show that EmbedLLM enhances accuracy and latency in model routing and can predict benchmark scores reliably. The embeddings also reflect intrinsic model attributes, useful for identifying task-specific strengths, even in models not explicitly trained for certain tasks.

**Strengths:**

- The paper innovatively proposes the embedding of LLMs to facilitate managing and comparing them.
- The experiments in the paper are comprehensive, tested on 112 large models

**Weaknesses:**

- The paper proposes a method for encoding LLMs. However, in the implementation, this encoding is merely based on model IDs, treating each model entirely as a black box. With only 30,000 data for training, can the resulting encoding truly capture all the characteristics of the models? Large models differ significantly in their strengths across various domains and capabilities. Can such an approach, based solely on one round question-answer pairs, truly distinguish the models’ abilities when facing complex reasoning problems? Another issue is whether the scale of the proposed embedding network is sufficient to represent the characteristics of numerous models effectively.

- A little confused about line 256-257. Why are two values (p(m,q)_0 and p(m,q)_1) output here? Is it simply to add a nonlinear operation?

- The experiments do not appear to categorize models by scale. Intuitively, the larger the model, the smaller the performance differences between models. What is the authors' view on this issue?

- The paper does not seem to provide a clear and intuitive illustration of the overall architecture, including training, input, and output processes. Figures 1-3 are quite similar and take up too much space, and it wasn’t until section 4.1 that I understood how the whole system works. Of course, I do not deny the paper's contributions, but I suggest the authors improve the visual presentation.

**Questions:**

See the weaknesses

---

> ### Author Response · Authors · 2024-11-21
> **Rebuttal Response (1/1)**
>
> We thank the reviewer for their comments and for taking the time to review our paper. We also appreciate their comments about finding our work innovative. We respond to some of the reviewer’s comments below:
>
> W1: “With only 30,000 data for training, can the resulting encoding truly capture all the characteristics of the models?”
>
> We would like to clarify that our current embedder certainly does not capture ALL characteristics of the models. Intuitively, if the model behavior data has zero information about how models perform in complex reasoning tasks, then we argue that no method could effectively predict model behaviors on them. However, we would like to emphasize that the key contribution of our work is that **our framework could effectively capture and compress whatever information that is available in our dataset** in a compact embedding form, and our experiment in correctness forecasting as well as model routing suggests that the **model embeddings are powerful summarizers** of model strengths and weaknesses within the range of model behaviors that is tested in our selected benchmarks. We believe that our framework could also extend on complex reasoning tasks if we're able to at least reconstruct some of the relevant tasks.
>
> W2: “Why are two values (p(m,q)_0 and p(m,q)_1) output here? Is it simply to add a nonlinear operation?”
>
> The decoder can be seen as a binary classifier so the two outputs are the two logits for classifying to label 0 or 1, and we combine them into one logit by taking their difference.
>
> W3.1: “The experiments do not appear to categorize models by scale.”
>
> In fact in figure 6 we performed some analysis on scale that indicates our embedding intrinsically captures the scale of the model (which is likely to be represented by the overall ability across all benchmarks). Specifically, we demonstrated that for all models of size 7B, 13B, or 70B, **their embeddings’ intra-community distance is smaller than inter-community distance, indicating a presence of clusters**, even though the data doesn’t explicitly contain the size information of the models.
>
> W3.2: “Intuitively, the larger the model, the smaller the performance differences between models. What is the authors' view on this issue?”
>
> We respectfully disagree with the statement that the performance gap between larger models is smaller. While many large models share overlapping training data, recent developments focus on specialization through domain-specific fine-tuning, or utilizing various feedback like program compiler for code generation or theorem verifiers for math. This led to remarkable divergences of performance even among similarly scaled models. We believe that larger models are exhibiting more and more distinctive strengths and weaknesses specific to certain tasks rather than converging in performance, and so our method could be a sustainable solution in this future we envision.
>
> W4: “The paper does not seem to provide a clear and intuitive illustration of the overall architecture, including training, input, and output processes.”
>
> We thank the reviewer for this suggestion. We will refine the illustrations and add a figure that explains our architecture explicitly.
>
> Finally, we would like to thank the reviewer once more for their comments which we find to be really constructive. If the reviewer finds our responses above to answer their questions, we gently request the reviewer to take them into account before finalizing their reviews and revise their scores as they find appropriate.

---

> > ### Comment · Reviewer_Gpg4 · 2024-11-27
> >
> > Thanks for the response. I am OK about it, and will raise my score based on the clarification。

---

> > > ### Author Response · Authors · 2024-11-27
> > >
> > > Thank you for taking the time to carefully review our responses and for reconsidering your score. We sincerely appreciate your thoughtful engagement with our work and your constructive feedback throughout the review process!

---

### Official Review · Reviewer_WPNj · 2024-11-04

**Soundness:** 2
**Presentation:** 2
**Contribution:** 3
**Rating:** 6
**Confidence:** 3

**Summary:**

This paper proposes a novel framework for generating compact vector representations of LLMs to enhance model routing, task efficiency, and performance forecasting. The EmbedLLM framework creates embeddings that capture important characteristics of different LLMs, such as suitability for specific tasks like coding or conversational response generation. Experiments results indicate that the embeddings effectively capture key characteristics of LLMs, enabling efficient and accurate task allocation and performance prediction across a variety of benchmarks.

**Strengths:**

Embedding LLMs to handle downstream tasks is indeed a fascinating approach! This method allows you to create compact representations of each model that capture its unique strengths and weaknesses, enabling efficient task-specific decisions without running each model on every input. This approach streamlines the workflow significantly, as it allows for general-purpose embeddings that can adapt to a variety of downstream tasks without retraining the models themselves. It's especially beneficial in settings where computational resources are a concern or when the model pool is large.

**Weaknesses:**

The term "decoder" in this paper is a bit misleading. In typical encoder-decoder architectures, the "decoder" reconstructs or generates the output in its full or intended form, such as reconstructing text in sequence-to-sequence tasks. Here, however, the so-called "decoder" is merely a binary classifier that outputs a label indicating whether the LLM correctly answered a question.
We have to re-train the embedder if we want to represent new models, this makes the whole framework non-scalable. I'd like to see details cost metrics for re-train embedder for new model vs traditional benchmarking approaches.

**Questions:**

Please address the weakness I mentioned above.

---

> ### Author Response · Authors · 2024-11-21
> **Rebuttal Response (1/1)**
>
> We thank the reviewer for their comments and for taking the time to review our paper. We also appreciate their comments about finding our work novel and effective. We respond to some of the reviewer’s comments below:
>
> W1: “The term "decoder" in this paper is a bit misleading.”
>
> We thank the reviewers for pointing out this possible confusion. In the specific setting of our binary correctness dataset, the decoder is indeed a binary classifier, and the reason why we call it a decoder is because it is used to reconstruct every entry of our correctness matrix. We will place more emphasis in explaining this in the updated writing to make it less ambiguous.
>
> Nonetheless, we re-emphasize that **our framework is generalizable to learning from other formats of data**, for instance we could learn to reconstruct exact model responses to questions and in that setting the decoder would be reconstructing text as the reviewer suggested. In this work, the main focus is to convey that we can employ such encoder-decoder architecture to learn compact representations of various LLMs that facilitate downstream tasks, given some data of model behaviors.
>
> W2: “I'd like to see details cost metrics for re-train embedder for new model vs traditional benchmarking approaches.”
>
> About the retraining cost, we would like to reassure the reviewer that training our embedder **takes a trivial amount of computation compared to traditional benchmarking approaches**. Below is some experimental details:
>
> Using fvcore package to perform flop analysis, we performed an example on a training set leaving out all questions from MMLU, which results in a correctness matrix of around 20,000 questions on 112 models. Training our embedder on this dataset for 50 epochs using a batch size of 2048 results in peak memory usage of less than 1GB and costs 107.71 TFlops in total. This is **approximately equivalent to the cost of running 60 inferences of length 128 on a 7B model** (calculation referring https://arxiv.org/pdf/2001.08361). That is, benchmarking all 112 models on MMLU test set (about 14,000 questions) will be at least (as we have larger models) 112 * 14000 / 60 = **26000 times more expensive**. Hence, we believe that our framework could effectively leverage the extensive demand on performing traditional benchmarking.
>
> Finally, we would like to thank the reviewer once more for their comments which we find to be really constructive. If the reviewer finds our responses above to answer their questions, we gently request the reviewer to take them into account before finalizing their reviews and revise their scores as they find appropriate.

---

> > ### Comment · Reviewer_WPNj · 2024-11-26
> >
> > Thank you for you response. It addressed my concerns. I will increase my rating to 6.

---

> > > ### Author Response · Authors · 2024-11-27
> > >
> > > Thank you for taking the time to carefully review our responses and for reconsidering your score. We sincerely appreciate your thoughtful engagement with our work and your constructive feedback throughout the review process!

---

### Author Response · Authors · 2024-11-23
**Rebuttal Reminder**

Dear Reviewer,

We wanted to kindly remind you about the discussion phase for our paper, as your engagement and potential revision of the score would greatly contribute to the evaluation process. Please let us know if there’s any clarification or additional information we can provide to assist you.

Thank you for your time and valuable feedback!

---

### Meta-Review · Area_Chair_8pDX · 2024-12-22

**Metareview:**

This paper presents an approach for learning a representation of the language model which helps it produce an embedding for a more compact representation of the language model. This allows the embedding to be used in three different tasks of interest: model routing, accuracy prediction and binary correctness prediction that bypasses running inference over a large set of benchmark tasks. Embeddings are learned through matrix factorization, where the matrix represents the performance of language models on test data points. The paper presents experiments for accuracy and latency in model routing and benchmark performance prediction. The authors claim that embeddings could reflect intrinsic model attributes, useful for identifying task-specific strengths, even in models not explicitly trained for certain tasks.


**Strengths:** The paper presents an innovative approach to streamline the evaluation of LLMs by proposing compact representations. Experiments over a large number of models are presented.

**Weaknesses:** It is natural to be concerned about the strength of the learned representations, where a single embedding is supposed to represent an entire language model. This raises questions about how well the models may generalize to new settings (other tasks), models and evaluation points.

**Reason for acceptance (poster)**: See strengths above. Despite the obvious weakness about representational capacity, the work shows improvements on the benchmarks of choice. Compact representations can be useful for a variety of use cases beyond the ones addressed in this work.

**Additional Comments On Reviewer Discussion:**

Reviewers raised questions about the use of a decoder model for classification tasks, authors addressed this by asserting the generalizability of the decoder, however no experiments were presented that showcased generative tasks in these models. There were other questions about details in the paper, which were also clarified in the discussion. However, there are some outstanding questions about the capacity of the learned embeddings which were not adequately addressed by the authors. One reviewer’s concern about the qualitative evaluation of the embedding was not adequately addressed which resulted in the reviewer reluctantly increasing their score to a 8, even though they would have liked to award a score of 7 to this work.

---

### Decision · Program_Chairs · 2025-01-22

Accept (Spotlight)